# Socioeconomic predictors of COVID-19-related health disparities among United States workers: A structural equation modeling study

**Ariadna Capasso**, **Sooyoung Kim**, **Shahmir H. Ali**, **Abbey M. Jones**, **Ralph J. DiClemente**, **Yesim Tozan**\*

School of Global Public Health, New York University, New York, New York, United States of America

\* tozan@nyu.edu

## Abstract

The COVID-19 pandemic has disproportionately impacted the physical and mental health, and the economic stability, of specific population subgroups in different ways, deepening existing disparities. Essential workers have faced the greatest risk of exposure to COVID-19; women have been burdened by caretaking responsibilities; and rural residents have experienced healthcare access barriers. Each of these factors did not occur on their own. While most research has so far focused on individual factors related to COVID-19 disparities, few have explored the complex relationships between the multiple components of COVID-19 vulnerabilities. Using structural equation modeling on a sample of United States (U.S.) workers (N = 2800), we aimed to 1) identify factor clusters that make up specific COVID-19 vulnerabilities, and 2) explore how these vulnerabilities affected specific subgroups, specifically essential workers, women and rural residents. We identified 3 COVID-19 vulnerabilities: financial, mental health, and healthcare access; 9 out of 10 respondents experienced one; 15% reported all three. Essential workers [standardized coefficient (β) = 0.23; unstandardized coefficient (B) = 0.21, 95% CI = 0.17, 0.24] and rural residents (β = 0.13; B = 0.12, 95% CI = 0.09, 0.16) experienced more financial vulnerability than non-essential workers and non-rural residents, respectively. Women (β = 0.22; B = 0.65, 95% CI = 0.65, 0.74) experienced worse mental health than men; whereas essential workers reported better mental health (β = -0.08; B = -0.25, 95% CI = -0.38, -0.13) than other workers. Rural residents (β = 0.09; B = 0.15, 95% CI = 0.07, 0.24) experienced more healthcare access barriers than non-rural residents. Findings highlight how interrelated financial, mental health, and healthcare access vulnerabilities contribute to the disproportionate COVID-19-related burden among U.S. workers. Policies to secure employment conditions, including fixed income and paid sick leave, are urgently needed to mitigate pandemic-associated disparities.

**Data Availability Statement:** Data were made publicly available via ICPSR COVID-19 Data

Repository Project (number openICPSR-120308, https://doi.org/10.3886/E120308V1).

**Funding:** The authors received no specific funding for this work.

**Competing interests:** The authors have declared that no competing interests exist.

## Introduction

The coronavirus disease (COVID-19) pandemic has claimed a devastating toll on people's physical and mental health [1, 2] and caused social and economic hardship both globally and in the United States (U.S.) [3]. In the U.S., surveys have found that 40% of people experienced poor mental health during the pandemic [4], 15% lost their jobs [3], one in seven experienced food insecurity [5, 6], and over 7% of adults experienced housing insecurity [7] during the pandemic. Poor mental health outcomes have been associated with heightened risk of exposure to COVID-19 [8] and financial instability [9].

The COVID-19 pandemic has disproportionately affected the working poor and exacerbated existing socioeconomic and health disparities. Essential workers, which include frontline healthcare workers and others who must perform work-related duties onsite (e.g., first responders, grocery store workers, agricultural workers) [10], are at increased risk for severe acute respiratory syndrome coronavirus 2 (SARS-CoV-2) infection, and also face a significant burden of economic insecurity and job precarity (e.g., low wages, variable income, no paid sick leave, no health insurance), which could lead to poor mental health [4, 11].

Women have also borne the brunt of the pandemic and associated prevention measures. First, women make up a majority of essential workers in certain sectors; for example, women make up 76% of healthcare workers [12]. Second, women, who make on average lower wages than men, are disproportionately employed in the informal economy and experienced greater job and income loss during the pandemic, further widening the pay gap [13]. Third, women's share of child- and elder-care burden has increased during the pandemic due to school closures and lockdowns [14]. In addition, the overall prevalence of mental illness is higher among women (24.5%) than among men (16.3%) [15]; economic insecurity and increased care burden may contribute to the worsening of sex-based mental health disparities [16].

COVID-19 affected rural and urban areas in drastically different ways [17, 18]. For example, the disease spread more rapidly in urban areas, but lockdown measures may have exacerbated social isolation and loneliness for rural residents with already limited social networks [19]. During the COVID-19 pandemic, rural populations experienced higher rates of mental health problems than urban populations [18]. COVID-19 may have exacerbated historical access to care barriers (due to, e.g., lower provider-to-population ratio, longer distances to care) for rural residents [20].

Many social and economic factors affected how COVID-19 affected the health and well-being of the U.S. population. These factors were often interconnected and co-occurring, or 'clustered' among specific population subgroups. Understanding these complex relationships calls for a systems-thinking approach. Systems thinking presents a holistic approach to understand the relationships and interdependence between multiple phenomena, considering the multi-level factors that contribute to the spread of and mitigation or containment of disease as part of dynamic systems [21]. Since the beginning of COVID-19, several papers have used systems thinking to tackle issues as diverse as testing-site deployment [22], and emergency communications [23]. However, few studies have applied this approach to understanding the myriad factors contributing to systemic COVID-19 vulnerabilities [24, 25]. The phenomenon of clustering of behavioral, social and economic risk factors that lead to disease has long been studied to understand population-level health disparities [26, 27]. For the purposes of this study, we define vulnerability as a cluster of closely correlated factors that place specific population subgroups at greater risk for adverse COVID-19-associated outcomes. We consider eleven socioeconomic factors in the model.

This study sought to map the relationships between sociodemographic, economic, and health-related factors that constitute different patterns of vulnerabilities that emerged during

the COVID-19 pandemic in the U.S. This study aimed to 1) identify the cluster of factors that make up specific vulnerabilities associated with COVID-19 exposure risk and 2) explore how these vulnerabilities intersected and differentially affected subpopulations, specifically women, essential workers, and rural versus urban residents.

## Materials and methods

### Participant recruitment and survey administration

This is an observational study based on an analysis of survey data collected by the researchers in the early days of the COVID-19 pandemic in the U.S. Study procedures have been detailed elsewhere [28]. Briefly, an advertisement campaign on Facebook and affiliated platforms was used to distribute a link to an anonymous English-language survey programmed on Qualtrics (Provo, UT) from April 16–21, 2020, in the initial phase of the outbreak in the U.S. The survey's first two questions assessed eligibility criteria: being an adult (aged $\geq$ 18 years) and residing in the U.S. No participation incentives were given. The analytic sample includes 2845 respondents who answered the question on current employment status as full-time employee (n = 2067), part-time employee (n = 402), self-employed (n = 371) or military personnel (n = 5). Respondents who were not in paid employment (e.g., unemployed) were excluded (n = 2217). In sum, this is a convenience sample of English-speaking social media users in the U.S. The New York University Institutional Review Board reviewed and approved study procedures as exempt and waived the requirement for explicit written or oral consent.

### Survey measures

Survey questions were informed by the World Health Organization tool for behavioral insights on COVID-19 [29] and the Kaiser Family Foundation Coronavirus Poll [30]. Respondents' sociodemographic information included sex, age (18–39 years, 40–59 years, $\geq$ 60 years), race [due to very low numbers this variable was recoded as dichotomous: non-Hispanic white, Black, Indigenous, People of Color (BIPOC)], highest educational attainment ($\geq$ college degree, $\leq$some college), marital status (single/separated/divorced; married/partnered), and type of residence area (urban, suburban, rural). In the model, we used two dummy variables for the residence question: urban (versus suburban and rural combined) and rural (versus suburban and urban combined). Being an essential worker was assessed by the question, "Are you considered an essential worker (i.e., do you have to go into work when others in your community have been asked to stay at home)?" (yes/no).

Financial status indicators included: 1) annual household income ($\geq$ $90,000; $50,000 - <$ $90,000; $<$ $50,000); 2); variable income (in response to "How do you get paid?," salaried employees were coded as fixed income and those who were paid per hour, per job or based on other arrangements as variable income); 3) lost income due to COVID-19 (yes/no); 4) past 3-months food insecurity (assessed with the six-item United States Department of Agriculture Household Food Security Survey Module [31]; scores $<$ 2 = not food insecure; 2–6 = food insecure); 5) whether a respondent could not afford to self-quarantine if mandated to do so (yes/no); and 6) ability to perform work from home in response to the question, "If you were required to remain at home because of a quarantine or work closure, would you be able to do at least part of your job from home?" (yes/no).

Anxiety and depressive symptomatology due to COVID-19 were assessed with an adapted brief version of the Patient Health Questionnaire (PHQ-4) [32, 33]; two items measured anxiety and two depressive symptomatology. We used the following modified stem question, "Over the last 7 days, how often have you been bothered by any of the following problems because of the Coronavirus outbreak?" Responses were rated on a 4-point Likert-type scale

ranging from "Not at all" (0) to "Nearly every day" (3). Summed scores ranged from 0 to 6 on each subscale, with higher scores indicative of higher symptomatology. The subscales demonstrated acceptable internal reliability (Spearman-Brown reliability estimate $\rho = 0.74$ and $\rho = 0.79$ for anxiety and depression, respectively). Variables were also dichotomized by the clinical cutoff ($\geq 3$) to assess the prevalence of anxiety and depression.

Traumatic stress symptoms during COVID-19 were assessed with the Impact of Events Scale-6 (IES-6) [34, 35]. Sample modified items were, "I thought about Coronavirus when I didn't mean to" and "I tried not to think about Coronavirus." Items were rated on a 5-point Likert scale ranging from "Not at all" (0) to "Extremely" (4). Higher mean scores were indicative of a higher mental health COVID-19-associated burden. The scale's reliability coefficient was $\alpha = 0.84$. The scale was also dichotomized at the clinically significant cutoff of 1.75, which is indicative of traumatic stress [36].

Healthcare access was measured by 1) the employer offers no paid sick leave (yes/no) and 2) the respondent has no health insurance (yes/no).

## Statistical analysis

After examining the distribution and frequencies of the variables of interest, we used structural equation modeling (SEM) to evaluate a hypothesized conceptual model of COVID-19-related vulnerabilities and to assess clustering of eleven socioeconomic factors associated with these vulnerabilities (Fig 1). Stata 15.1 (Stata Corporation LP, College Station, TX) was used to conduct univariate analysis and Mplus 8.3 (Muthén & Muthén, 2017) to conduct the SEM.

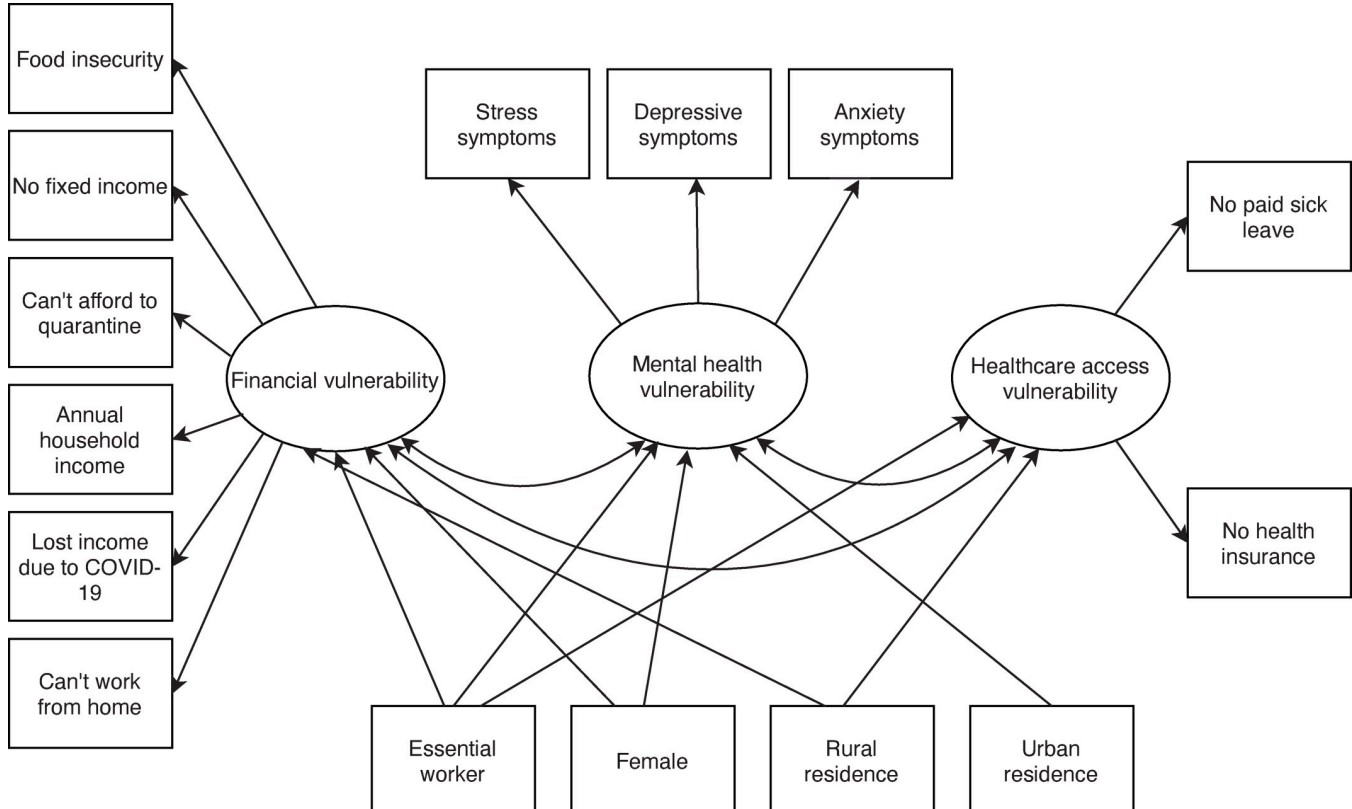

**Fig 1. Conceptual model depicting the relationship of sex, essential worker and residence status to financial, mental, and healthcare access vulnerabilities during the COVID-19 pandemic among U.S. workers.**

## Model estimation and model fit

Parameters and 95% confidence intervals were estimated and tested using a weighted least-squares estimator (option WLSMV in Mplus). Given the variation in the spread of COVID-19 in the U.S. and the differences in state-level response to the pandemic, state clustering was accounted for in the model. Global model fit was assessed based on the chi-square test of exact fit, the Comparative Fit Index (CFI) [37, 38], the Root Mean Square Error of Approximation (RMSEA), and the Standardized Root Mean-Squared Residual (SRMR). CFI values of $\geq 0.95$, RMSEA values $\leq 0.06$ and SRMR values $\leq 0.08$ were considered to determine appropriate model fit [39]. Modification indices were used to test localized fit. Model identification entailed an iterative process guided by the literature and an examination of the modification indices whereby a limited number of modifications were incorporated into the model until the satisfactory fit was achieved. One of the expectations of statistical models for use in the field of public health is for a model's utility to extend beyond the data employed to develop and test it [40]. This process often entails the development of a theory-based model using one sample and then validating the model in another sample [41]. However, this is at times not possible. To test the robustness of our model, we randomly split the sample in halves and assessed if the data fit the model adequately in each subsample, and if the direction and significance of the coefficients remained stable [41].

Four variables had missing data (lost income due to COVID-19 and ability to work from home = 2.6% missing values; no paid sick leave = 10%; and annual household income = 12.3%). Because of missing values, the model's sample size reduced from 2,845 to 2,800. Full-information maximum likelihood (FIML) as implemented in Mplus was used to address missing data [42]. We reported standardized coefficients (β) and unstandardized coefficients (B) with 95% confidence intervals for each parameter estimated.

## Results

### Univariate analysis

As seen in Table 1, a majority of respondents were women (56.2%), aged 40 to 59 years (56.5%), non-Hispanic white (93.0%), with a college degree (60.1%), married or partnered (73.8%), and suburban residents (55.7%). Respondents resided in the 50 U.S. states, the District of Columbia, and Puerto Rico.

Over one fifth of the sample (21.5%) earned an annual household income below $50,000; over half had variable income (55.4%); 39.6% had lost income due to COVID-19; 12.3% were food-insecure; 28.9% could not afford to quarantine; and 39.9% could not work from home.

Mean depressive symptoms were 1.7 on a scale of 6 [standard deviation (SD) = 1.9], mean anxiety symptoms 2.4 on a scale of 6 (SD = 2.1), and traumatic stress 1.8 on a scale of 4 (SD = 1.1). When examining the clinical cutoffs, 25.9% of respondents had scores consistent with possible depression, 40.3% with possible anxiety, and 51.2% with traumatic stress.

Over a quarter of respondents did not have paid sick leave (27.1%) and 4.9% did not have health insurance. When examining the cumulative distribution of factors, a majority of respondents experienced 3 factors (19%), with most respondents endorsing between 1 (13%) and 5 (12%) factors (See Fig 2).

### Structural equation models

The fit of the initial SEM, $\chi^2 = 879.42$, degrees of freedom (DF) = 73, p < 0.001; CFI = 0.83, RMSEA = 0.06, and SRMR = 0.07 indicated that the data did not fit the model satisfactorily. Guided by our understanding of the relationship between the variables and by the

**Table 1. Descriptive characteristics of working adults during the COVID-19 pandemic in the US, N = 2,845, April 2020.**

| | n (%) | Mean (SD) |
|---|---|---|
| *SOCIO-DEMOGRAPHIC CHARACTERISTICS* | | |
| **Female** | 1585 (56.2) | |
| **Age group** | | |
| 18–39 | 566 (19.9) | |
| 40–59 | 1607 (56.5) | |
| 60+ | 672 (23.6) | |
| **White race** | 2647 (93.0) | |
| **Some college or below** | 1127 (39.9) | 60.1 |
| **Married/partnered** | 2099 (73.8) | |
| **Residence type** | | |
| Rural | 830 (29.2) | |
| Suburban | 1584 (55.7) | |
| Urban | 431 (15.2) | |
| **Essential worker** | 1513 (53.6) | |
| *FINANCIAL FACTORS* | | |
| **Annual household income** | | |
| 90,000 and over | 1215 (48.4) | |
| 50,000-<90,000 | 759 (30.2) | |
| <50,000 | 537 (21.4) | |
| **No fixed income** | 1563 (55.4) | |
| **Lost income due to COVID-19** | 1116 (39.6) | |
| **Is food insecure** | 349 (12.3) | |
| **Can't afford to quarantine** | 821 (28.9) | |
| **Can't work from home, n (%)** | 1116 (39.9) | |
| *MENTAL HEALTH FACTORS* | | |
| **Depressive symptoms (range: 0–6)** | | 1.7 (1.9) |
| **Anxiety symptoms (range: 0–6)** | | 2.4 (2.1) |
| **Stress symptoms (Range: 0–4)** | | 1.8 (1.1) |
| *HEALTHCARE ACCESS FACTORS* | | |
| **Employer does not offer paid sick leave** | 701 (27.1) | |
| **No health insurance** | 139 (4.9) | |

modification indices, we incorporated several modifications until adequate global model fit was achieved: $\chi^2$ = 504.07, DF = 65, p < 0.001; CFI = 0.91, RMSEA = 0.05, and SRMR = 0.06. Fig 3 presents the final SEM with standardized path coefficients (covariances and correlations are omitted for simplicity). S1 Table presents the fit statistics for each model, and Table 2 displays the final model's estimated parameters.

## Vulnerabilities analysis

Three vulnerabilities emerged from the analysis: financial, mental health, and access to healthcare. Essential workers were more financially vulnerable than non-essential workers (β = 0.23; B = 0.21, 95% CI = 0.17, 0.24); however, they reported better mental health (β = -0.08; B = -0.25, 95% CI = -0.38, -0.13) and experienced fewer healthcare access barriers (β = -0.06; B = -0.10, 95% CI = -0.18, -0.01) than other workers.

Women experienced worse mental health than men (β = 0.22; B = 0.65, 95% CI = 0.65, 0.74), but there were no significant differences in regard to finances or healthcare access. We

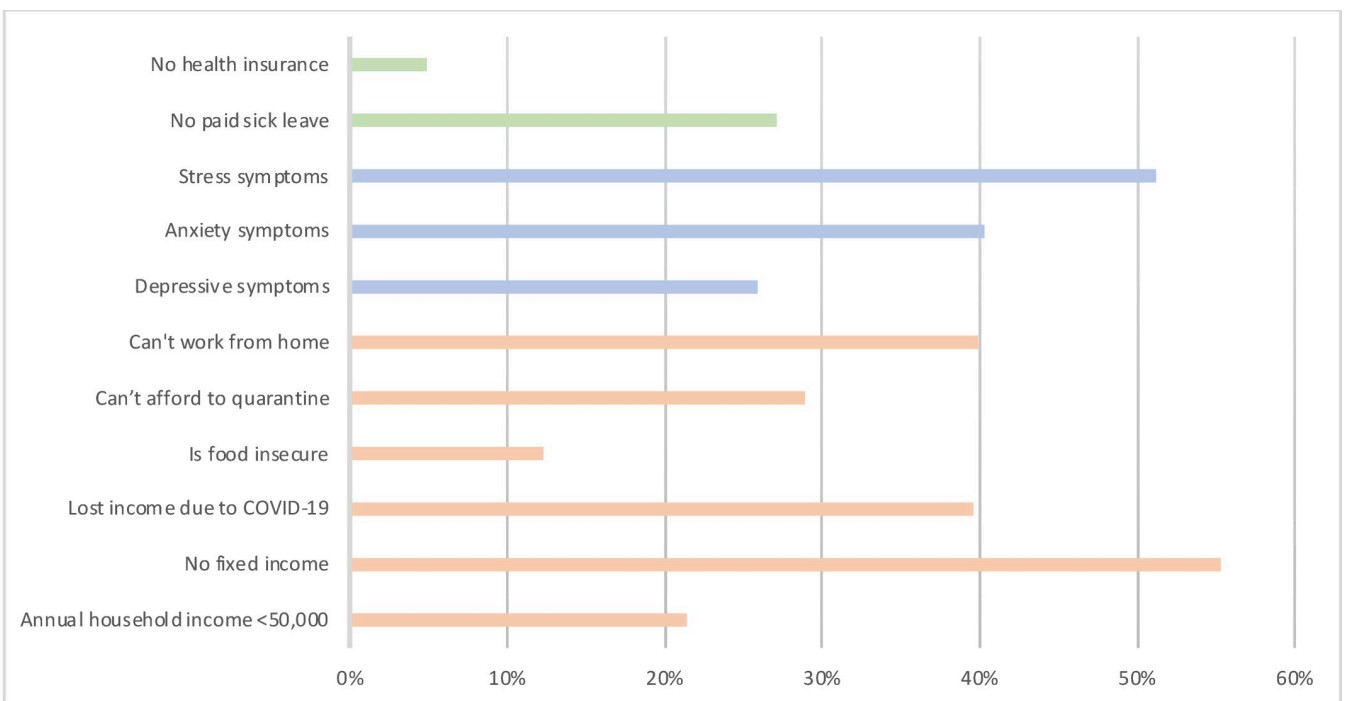

Legend

Healthcare access vulnerability
Mental health vulnerability
Financial vulnerability

**Fig 2. Cumulative distribution of factors among 2,845 U.S. workers during the COVID-19 pandemic (April 2020).**

found that urban residents experienced more financial (β = 0.07; B = 0.08, 95% CI = 0.05, 0.12) and mental health (β = 0.03; B = 0.14, 95% CI = 0.14, 0.25) vulnerabilities than non-urban residents. Rural residents were more financially vulnerable (β = 0.13; B = 0.12, 95% CI = 0.09, 0.16) and faced more healthcare access barriers (β = 0.09; B = 0.15, 95% CI = 0.07, 0.24) than non-rural residents, but experienced better mental health (β = -0.08; B = -0.27, 95% CI = -0.39, -0.16). Table 3 presents the distribution of vulnerabilities per the four key characteristics of interest. Only 10% of respondents did not have any vulnerabilities; 34% had 1 vulnerability, 41% had two vulnerabilities, and 15% reported all three.

## Post hoc analyses

We found evidence for the robustness of the model as both the model fit and the estimated coefficients were similar in the full and random split samples. The model fit the data equally well in the full and split samples (S2 Table), the signs of the coefficients in both split samples were consistent with those of the full model, and the significance level changed in only 4 of the 36 coefficients estimated (S3 Table).

## Discussion

Our analysis indicates that three distinct vulnerabilities emerged during the COVID-19 pandemic among U.S. working adults: financial, mental health, and access to care. By using a systems-thinking approach, we were able to better understand how U.S. workers were affected in

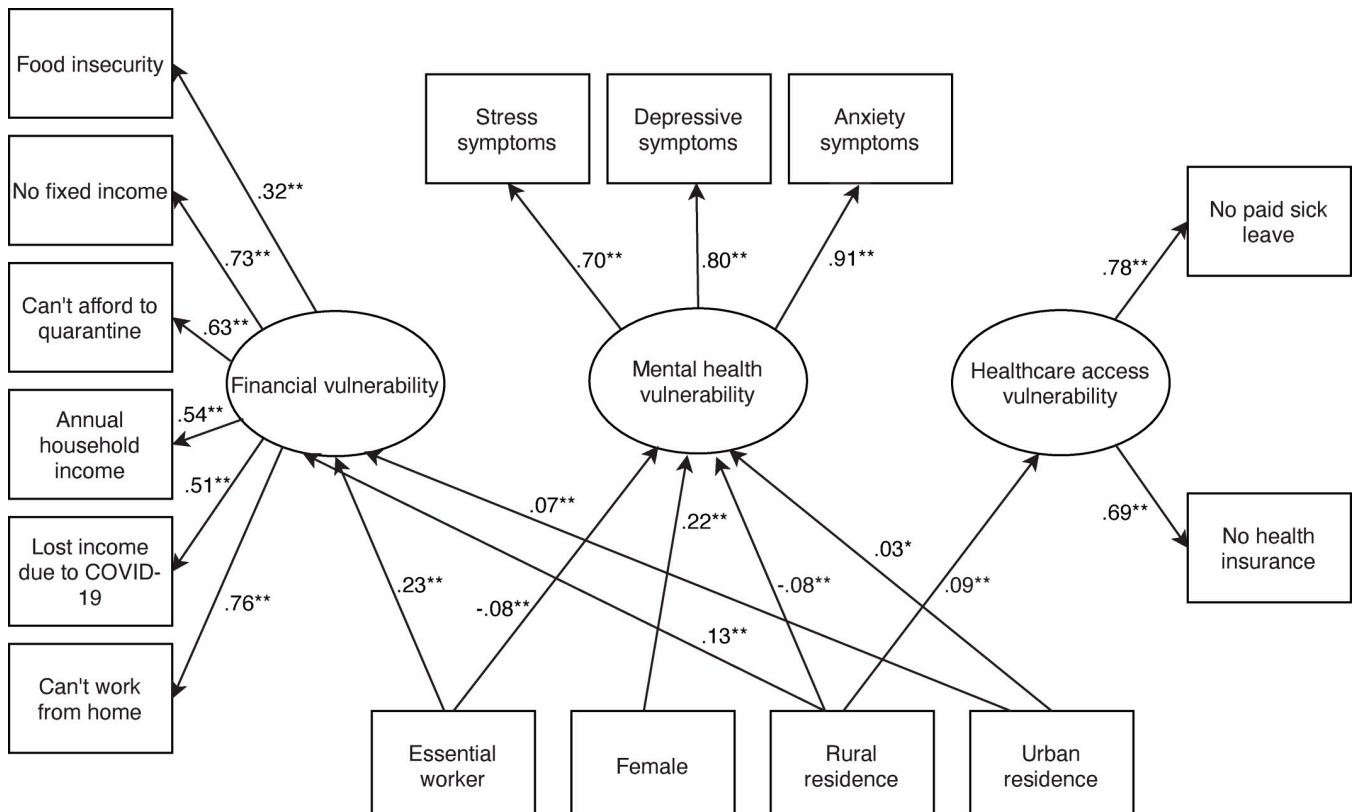

**Fig 3. Structural equation model depicting the relationship of sex, essential worker and residence status on financial, mental health, and access to care vulnerabilities among 2800 workers in the US during the COVID-19 pandemic (April 2020).** Notes: Standardized coefficients. $^*$p < .05; $^{**}$p < .001. r for mental health with financial vulnerability = .11, p < .001. r for healthcare access with financial vulnerability = .84, p < .001. r for healthcare access with mental health vulnerability = .06, p < .05. r for can't work from home and no fixed income = .34, p < .001. r for food insecurity with depression = .36, p < .001. r for food insecurity with anxiety = .35, p < .001. r for food insecurity with income = .21, p < .001. r for food insecurity with can't afford to quarantine = .32, p < .001. r for lost income due to COVID-19 and no paid sick leave = .49, p < .001.r for no fixed income with stress = -.26, p < .001. r for no fixed income with paid sick leave = .40, p < .001.

**Table 2. Final model paths: Standardized coefficients and unstandardized coefficients with 95% confidence intervals, N = 2,800, April 2020.**

| Outcome | Explanatory variable | β | B | 95% CI |
|---|---|---|---|---|
| Financial vulnerability | Essential worker | 0.23 | 0.21 | 0.17, 0.24 |
| | Female | 0.03 | 0.02 | -0.01, 0.06 |
| | Urban residence | 0.07 | 0.08 | 0.05, 0.12 |
| | Rural residence | 0.13 | 0.12 | 0.09, 0.16 |
| Mental health vulnerability | Essential worker | -0.08 | -0.25 | -0.38, -0.13 |
| | Female | 0.22 | 0.65 | 0.65, 0.74 |
| | Urban residence | 0.03 | 0.14 | 0.14, 0.25 |
| | Rural residence | -0.08 | -0.27 | -0.39, -0.16 |
| Healthcare access vulnerability | Essential worker | -0.06 | -0.10 | -0.18, -0.01 |
| | Female | -0.05 | -0.07 | -0.14, 0.02 |
| | Urban residence | 0.02 | 0.04 | -0.08, 0.16 |
| | Rural residence | 0.09 | 0.15 | 0.07, 0.24 |

*Note*: β = standardized coefficients B = unstandardized coefficients 95% CI = 95% confidence intervals.

**Table 3. Distribution of cumulative vulnerabilities by key characteristics.**

| # vulnerabilities | Total | Essential workers | Women* | Urban | Rural |
|---|---|---|---|---|---|
| 0 | 278 (9.8) | 121 (8.0) | 133 (8.4) | 36 (8.4) | 81 (9.8) |
| 1 | 970 (34.1) | 517 (34.2) | 519 (32.7) | 148 (34.3) | 287 (34.6) |
| 2 | 1159 (40.7) | 656 (43.4) | 667 (42.1) | 176 (40.8) | 330 (39.8) |
| 3 | 438 (15.4) | 219 (14.5) | 266 (16.8) | 71 (16.5) | 132 (15.9) |

*n = 2821 because of missing.

different ways by the COVID-19 pandemic. A large number of respondents in our survey was adversely impacted, with 90% endorsing at least one vulnerability, and 15% all three.

Our findings indicate that essential workers have experienced a more dire financial impact due to COVID-19 than non-essential workers. This is consistent with the extant literature showing that essential workers were more likely to lose income during the pandemic [43]. Even before the pandemic, they were generally earning less income and worked in more unstable conditions than non-essential workers [12, 44]. Lack of safety-net benefits, such as paid sick leave, is likely to exacerbate economic instability. These findings underscore the need to buttress labor policies to secure employment conditions, including fixed income in place of hourly or task-based wages, and mandatory paid sick leave. Essential workers were less likely to report poor mental health and barriers to healthcare access than non-essential workers. This unexpected finding could be explained by several reasons. First, essential workers are a diverse group of people working in different sectors and under different labor conditions [10], some with more stability and benefits than others. The binary variable could have masked the sector- or other subgroup-specific effects. Second, the Affordable Care Act (ACA) has dramatically increased the proportion of insured individuals in the U.S., including those with non-employer-provided insurance [45, 46], which may have mitigated access-to-care barriers for those with no health insurance benefits [47]. Third, it is possible that essential workers who were most affected by the COVID-19 pandemic were less likely to participate in this survey. Indeed, studies of non-response bias have documented better mental health among voluntary than mandatory survey respondents [48] and have shown better healthcare utilization, a proxy for access to care, among survey respondents than non-respondents [49].

In our survey, women reported worse mental health of men. This is consistent with studies of the prevalence of depression [50] and anxiety [51] showing that women experience mental health disorders at about double the rate than men. It was unexpected that there were no sex differences for the other vulnerabilities. One explanation could be that we restricted the sample to currently employed respondents. Thus, we could not assess if more women became unemployed during the pandemic. Further, the data were collected early in the pandemic, which could have provided insufficient lag time to assess its full economic impact [52].

Financial vulnerability was present among both rural and urban residents, but more pronounced among rural residents. This is consistent with chronic financial vulnerability in rural areas. Another explanation could be that the average financial effect of COVID-19 among the urban population was diluted by the nature of concentrated inequality, with pockets of extreme poverty and of extreme wealth [53]. The full economic impact of temporary migration patterns, with affluent urban residents fleeing to the suburbs, would not have been captured by our survey data from April [54].

Mental health vulnerability was observed only in urban centers, which could be explained by the direct risk of SARS-CoV-2 infection, financial hardship, and abrupt changes in daily

routines due to mitigation measures. Further, many urban residents live in small and crowded apartments with limited outdoor space, exacerbating mental health outcomes [55].

Rural residents appeared to face more healthcare-access barriers than non-rural respondents. These findings are consistent with prior knowledge that healthcare services are limited in these areas [20, 56]. Stay-at-home orders, travel restrictions, and school and work closures, coupled with fear of contagion, particularly in densely populated areas, meant that people spent more time at home [57]. The association between rural residence and lower mental health vulnerability could be partially explained by the availability of green spaces [58], as well as larger living spaces [55], which may have mitigated COVID-19-associated angst in the context of reduced mobility. It cannot be ruled out that mental health may have worsened in rural areas later on, with the implementation of restrictive measures [59].

## Strengths and limitations

Our study adopted a systems-thinking approach to understand the impact of the COVID-19 pandemic on socioeconomic vulnerabilities, enabling the analysis of dynamics between multiple complex and associated factors simultaneously. Our analytic strategy of clustering by the state of residence allowed us to account for the different dynamics of COVID-19 spread and mitigation measures stemming from state-specific policy choices. The convenience-sampling strategy leveraging social media platforms yielded a relatively large sample size (N = 2800). Most important, restricting our analysis to the working adult population in the U.S. made our findings directly relevant for policymakers when considering the strengthening of labor policies during and after the pandemic.

The current study was subject to several limitations. First, our sample consisted of a non-probability convenience sample of social media users, primarily on Facebook. While 70% of U.S. adults own a Facebook account, the share of social media use is not distributed equally among demographic subgroups. In particular, BIPOC are less likely to have access to broadband and social media than their white counterparts since they are more likely to experience some of the structural inequalities addressed in this paper, such as precarious employment conditions and unmet basic needs [60]. In addition, BIPOC and those with poor mental health may be more reluctant to participate in web-based surveys [61–63]. All responses were self-reported, thus prone to reporting biases. Although validated scales were utilized, sensitive topics, such as income and mental health, may have caused respondents to provide less accurate, but more socially desirable, answers. Lastly, caution is required to interpret our findings, as the pandemic continued to evolve after survey data was collected in April 2020, only a few weeks after the first pandemic restrictions were imposed in the U.S. For policymakers to accurately measure the impact of COVID-19 on socioeconomic vulnerabilities, research is warranted utilizing a nationally representative sample.

## Conclusion

Study findings underscore the importance of a systems thinking approach to understanding COVID-19-related disparities experienced by U.S. workers, and highlight how interrelated financial, mental health, and healthcare access vulnerabilities may be contributing to the disproportionate COVID-19-related burden experienced by specific subpopulations. As the U.S. prepares for post-pandemic recovery, it should provide equitable resource allocation to support the varying needs of the population at the core of its public policy agenda. To do this, policymakers must understand how clusters of factors that constitute distinct vulnerabilities have differently affected subpopulations. Policymakers at federal and state level should consider the adoption of labor policies to secure employment conditions, including fixed income and paid

sick leave, which will be critical to mitigating pandemic-associated disparities. The strengthening of labor policies may be crucial in helping workers recover from the social and economic impact of the pandemic, and to mitigate such vulnerabilities in future crises.

## Supporting information

**S1 Table. Model fit indices.**
(XLSX)

**S2 Table. Model fit indices for the random split samples.**
(XLSX)

**S3 Table. Model paths: Unstandardized coefficients with 95% confidence intervals comparing the full and split sample models.**
(XLSX)

## Author Contributions

**Conceptualization:** Ariadna Capasso, Ralph J. DiClemente, Yesim Tozan.

**Data curation:** Ariadna Capasso, Shahmir H. Ali.

**Formal analysis:** Ariadna Capasso, Sooyoung Kim.

**Investigation:** Ariadna Capasso, Abbey M. Jones.

**Methodology:** Ariadna Capasso, Yesim Tozan.

**Resources:** Ralph J. DiClemente.

**Supervision:** Yesim Tozan.

**Writing – original draft:** Ariadna Capasso, Sooyoung Kim, Yesim Tozan.

**Writing – review & editing:** Shahmir H. Ali, Abbey M. Jones, Ralph J. DiClemente.

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
