## [Decision Letter · Decision Letter 0]

4 Nov 2021

PGPH-D-21-00803

Socioeconomic predictors of COVID-19-related health disparities among U.S. workers: A structural equation modeling study

Dear Dr. Tozan,

Thank you for submitting your manuscript to PLOS Global Public Health. After careful consideration, we feel that it has merit but does not fully meet PLOS Global Public Health’s publication criteria as it currently stands. Therefore, we invite you to submit a revised version of the manuscript that addresses the points raised during the review process.

In addition to the comments from the reviewers, please address the following points:

From https://journals.plos.org/globalpublichealth/s/submission-guidelines#loc-methods-software-databases-and-tools : Submissions to PLOS GPH presenting methods, software, databases, or tools (including models) must demonstrate that the new tool achieves its intended purpose. This requirement may be met by including a proof-of-principle experiment or analysis; if this is not possible, a discussion of the possible applications and some preliminary analysis may be sufficient.Would you please discuss how you might validate this model and what its possible applications are?58-69: please provide references for these assertions179: what was the cutoff for determining satisfactory fit using CFI/RMSEA/SRMR?206-208: this is a bit confusing – which factors are you referring to here? please clarify if it was 15% (as stated in the abstract) or 19% (as stated here) of respondents who experienced three factors; also please clarify the characterization of 19% as a majority and which 5 factors you’re referring to here given that in the abstract you refer to three factors. Or are you referring here to the characteristics listed in Table 1? Later you use the term “vulnerability” instead of “factor”.330-331: can you be more specific about where or which structural inequalities are discussed in the paper vis-à-vis BIPOC specifically (most discussion in the Introduction is about inequality by sex and SES)?338-9: is this a realistic recommendation at this point if not already carried out?

We look forward to receiving your revised manuscript.

Kind regards,

Karen D. Cowgill, PhD, MSc

Academic Editor

Journal Requirements:

1. Please provide separate figure files in .tif or .eps format only, and remove any figures embedded in your manuscript file.  If you are using LaTeX, you do not need to remove embedded figures.

2. Please note that your Data Availability Statement is currently missing the DOI/accession number of each dataset OR a direct link to access each database. If your manuscript is accepted for publication, you will be asked to provide these details on a very short timeline. We therefore suggest that you provide this information now, though we will not hold up the peer review process if you are unable.

Additional Editor Comments:

Copyediting suggestions by line in original submission (note that PLOS GPH does not provide copyediting services for manuscripts, so please review your manuscript carefully – the suggestions below are not meant to be burdensome but rather to improve the manuscript’s readability)

Title: write out U.S.

32: replace comma with a colon

33: please define beta and B in the abstract

37: as Reviewer 2 pointed out, mental health results are missing from the abstract

42: write out “coronavirus disease” prior to using the abbreviation COVID for the first time

43: delete comma, delete hyphen

45: please rewrite this sentence to avoid potential confusion – as written, “It” could be interpreted as referring to the survey rather than the pandemic

47: suggest specifying COVID-19 exposure risk

53 and 55 and 73 elsewhere: delete “etc.”

54: write out SARS-CoV-2 full name before using the abbreviation

77: insert hyphen: “calls for a systems-thinking approach”

81: insert hyphen: “testing-site deployment”; delete comma

82: if there are any such studies, please cite them here; if there are none that you’re aware of, modify the sentence to state that

86: delete hyphen

87: insert hyphen: “adverse COVID-19-associated outcomes”

91: delete colon

92: delete comma

105: delete colon

107: please include the name of the institution in the revised submission (note that this review is single-blind and so the authors’ names and institutions are already visible to reviewers)

114: delete colon; change semicolon following “18-39 years” to comma

115-6: change square brackets to parentheses and write out BIPOC

116-118, 124: replace semicolons in parentheses with commas

124-5 and 142: add space on either side of > or <

128: write out USDA

128-9, 182: add space on either side of = sign

131: add comma after “question”

135: delete comma

150: delete hyphen

153: delete colon and semicolon

Figures 1 and 2 captions: please add “among U.S. workers”

183: suggest stating “the model’s sample size was reduced from 2845 to 2800”

190: add comma after “Table 1”

196: delete hyphen

197-8: add hyphen “food-insecure”

200-201: remind the reader what the total possible score was (i.e., 1.7 on a scale of 6)

205: delete comma

226: “mental health”

Table 2 and Table 3 title: add population, N, date as in Table 1

243: replace “less” with “fewer”

252: add -d: “experienced”

254-5: keep semicolon after “vulnerabilities” but replace other semicolons with commas

259: add comma after “mental health”, hyphenate “systems-thinking” here where it modifies “approach”

268: hyphenate “safety-net”; add “leave” after “paid sick”

270: comma after “wages”

277: insert “those with” after “including”

278: hyphenate “access-to-care barriers”

281: delete comma

287: replace “than” with “of”

299: it may be helpful to remind readers that this was only a few weeks after the first pandemic restrictions were imposed in the US (also at line 102)

301: move “only” – “was observed only in urban centers”

306: hyphenate “healthcare-access barriers”

308: delete “the” before “lower mental health”

310-312: it may be helpful to provide more context here about the extent and timing of restrictions (in most cases these were not true lockdowns)

316: hyphenate “systems-thinking approach”

320: hyphenate “convenience-sampling strategy”

322: change “importantly” to “important”

330-1: comma after “counterparts”

342: hyphenate “systems-thinking approach”

345-7: suggest rewording these two sentences – perhaps “As the US prepares for post-pandemic recovery, it should put equitable resource allocation to support the varying needs of the population at the core of its public policy agenda. To do this, policymakers must …”

349: delete hyphen

References: please capitalize COVID and proper nouns (e.g., United States) in the reference list

Reviewers' comments:

Reviewer's Responses to Questions

**Comments to the Author**

1. Does this manuscript meet PLOS Global Public Health’s publication criteria? Is the manuscript technically sound, and do the data support the conclusions? The manuscript must describe methodologically and ethically rigorous research with conclusions that are appropriately drawn based on the data presented.

Reviewer #1: Yes

Reviewer #2: Yes

2. Has the statistical analysis been performed appropriately and rigorously?

Reviewer #1: Yes

Reviewer #2: Yes

3. Have the authors made all data underlying the findings in their manuscript fully available (please refer to the Data Availability Statement at the start of the manuscript PDF file)?

Reviewer #1: Yes

Reviewer #2: Yes

4. Is the manuscript presented in an intelligible fashion and written in standard English?

Reviewer #1: Yes

Reviewer #2: Yes

5. Review Comments to the Author

Reviewer #1: The theme is relevant and current. The study is well conducted and has practical application.

The authors present the research problem.

The goals are well defined. The results respond to the objective.

In the method, I suggest mentioning the type of study. The authors present the reference of the original study (reference 21). However, I suggest presenting some details of this study for better understanding.

Variables are clearly described.

The results and discussion are clear.

conclusions that are appropriately drawn based on the data presented.

Reviewer #2: The manuscript by Capasso et al., seeks to identify the socioeconomic predictors of COVID-19 related health disparities among US workers. The manuscript addresses a major concern as results obtained could be very important in shaping policy. Though the manuscript is properly written and the methods and results properly crafted, they are a few concerns that need to be addressed before the publication.

Listed below are few concerns I have with the manuscript:

1. Generally, I believe that the manuscript needs to be looked at a little more keenly and minor grammatical errors addressed. For instance, the last sentence in line 63 should begin with a conjunction like ‘In additional, moreover, etc. Lines 68 needs to rephrased for clarity. In addition, Line 75 – 76 should also be rephrased.

2. To provide context, it is important for the abstract to begin with a brief background that highlights the burden of the disease being addressed.

3. The abstract lacks the results related to the mental health status of the respondents

4. In the introduction, (line 42 -43), the authors should rather indicate the global burden exerted by COVID-19 and then talk about its impact on the US. The current statement is misleading as it seems to imply that COVID-19 has impacted only the US. In addition, the economic impact mentioned in line 43 is repeated in line 45. This should be corrected.

5. Though the need to provide written or oral consent was waived, the authors should clearly indicate where ethical clearance was obtained from.

I believe the following concerns addressed with go a long way to improve the quality of the manuscript and increase its suitability for publication.

6. PLOS authors have the option to publish the peer review history of their article (what does this mean?). If published, this will include your full peer review and any attached files.

**Do you want your identity to be public for this peer review?** For information about this choice, including consent withdrawal, please see our Privacy Policy.

Reviewer #1: No

Reviewer #2: **Yes: **Robert Adamu Shey

---

## [Editor Report · Decision Letter 1]

16 Dec 2021

Socioeconomic  predictors of COVID-19-related health disparities among United States workers: A structural equation modeling study

PGPH-D-21-00803R1

Dear Dr. Tozan,

We're pleased to inform you that your manuscript has been judged scientifically suitable for publication and will be formally accepted for publication once it meets all outstanding technical requirements.

Within one week, you'll receive an e-mail detailing the required amendments. When these have been addressed, you'll receive a formal acceptance letter and your manuscript will be scheduled for publication.

An invoice for payment will follow shortly after the formal acceptance. To ensure an efficient process, please log into Editorial Manager at https://www.editorialmanager.com/pgph/ click the 'Update My Information' link at the top of the page, and double check that your user information is up-to-date. If you have any billing related questions, please contact our Author Billing department directly at authorbilling@plos.org.

Kind regards,

Karen D. Cowgill, PhD, MSc

Academic Editor